Journal of Machine Learning Research  (2025) 1-14     Submitted 6/25; Revised 8/25; Published

# Low-Rank Adaptations for increased Generalization in Foundation Model features

**Vilde Schulerud Bøe[1]**                                          VILDESBO@IFI.UIO.NO
*[1]DSB group, Department of Informatics, University of Oslo, Norway*

**Andreas Kleppe[1,2,3]**                                          ANDREKLE@IFI.UIO.NO
*[2]Institute for Cancer Genetics and Informatics, Oslo University Hospital, Norway*
*[3]Centre for Research-based Innovation Visual Intelligence, UiT The Arctic University of Norway, Norway*

**Sebastian Foersch[4]**                     SEBASTIAN.FOERSCH@UNIMEDIZIN-MAINZ.DE
*[4]Institute of Pathology, University Medical Center, Mainz, Germany*

**Daniel-Christoph Wagner[4]**       DANIEL-CHRISTOPH.WAGNER@UNIMEDIZIN-MAINZ.DE

**Lill-Tove Rasmussen Busund[5,6]**       LILL.TOVE.RASMUSSEN.BUSUND@UNN.NO
*[5]Department of Medical Biology, UiT The Arctic University of Norway, Tromsø, Norway*
*[6]Department of Clinical Pathology, University Hospital of North Norway, Tromsø, Norway*

**Adín Ramírez Rivera[1]**                                          ADINR@UIO.NO

**Editor:**

## Abstract

For foundation models (FMs) to truly advance computational pathology, they must deliver consistent and reliable predictions under diverse, unseen test conditions. Without such robustness, clinical trust and widespread adoption remain out of reach. Although many FMs for histopathology now exist, they have to our knowledge not been systematically tested for robustness by external researchers on independent datasets. In this study, we evaluate the robustness of foundation model features on three separate histopathology datasets and find that their performance drops on external data. Our analysis also reveals that these models often encode dataset-specific information, limiting their generalizability. To address this issue, we train a Weight-Decomposed Low-Rank Adaptation (DoRA) with strong data augmentations to improve feature robustness. Our experiments show that models trained with this adapter exhibit fewer signs of dataset-specific information and may generate more robust features across domains. These results highlight the need for robustness testing and encourage incorporating robustness considerations into the development, training, and tuning of FMs for histopathology. The code for this work will be available at https://github.com/dsb-ifi/DoRA-for-FM-robustness

**Keywords:**   Domain Generalization, Robustness, Domain Shift, Computational Pathology, Foundation Model, Low-Rank Adaptation

## 1 Introduction

Deep learning has had a big impact on computational pathology, enabling accurate models for tasks such as survival analysis, tissue classification, and tumor detection (Echle et al., 2021). However, ensuring that these models generalize across different domains remains

a challenge (Kleppe et al., 2021). In histopathology, domain shifts are often unavoidable due to differences in staining protocols, tissue preparation procedures, scanner hardware, or institutional practices (Macenko et al., 2009). A model trained on data from one setting may encounter significantly different input characteristics in another, which can result in very different feature representations and degraded performance at deployment (Kleppe, 2022). Such lack of generalization and robustness threatens model reliability in deployment and poses risks to clinical utility and patient safety (Van der Laak et al., 2021).

Foundation Models (FMs) have emerged as a promising approach in this context. These models are pre-trained on large, diverse datasets and typically use extensive data augmentation during training. FMs aim to capture broad, transferable representations, and are often expected be more robust and less susceptible to domain shifts (Zhai et al., 2022; Oquab et al., 2023). However, recent studies suggest that FMs in histopathology may not reliably generalize to out-of-distribution conditions, raising questions about their dependability in varied clinical environments (Vaidya et al., 2024; Kömen et al., 2024; de Jong et al., 2025). Nevertheless, FMs offer substantial potential because they are trained on huge image sources to represent much biological information in feature embeddings that are far more manageable than the gigapixel Whole-Slide Image (WSI) they represent. If they can also produce more robust features, they may serve as powerful backbones for downstream pathology tasks, reducing the need for extensive training, data, and annotation.

In this work, we evaluate the robustness of FMs in histopathology and explore a potential approach to improve the feature representations under domain shifts using Weight-Decomposed Low-Rank Adaptations. This method is particularly suitable for FMs, which are challenging to fully fine-tune due to their large size and the lack of public access to the huge image sources most FMs are trained on.

## 2 Background

During the past few years, several FMs for histopathology have been released. These models are typically large Vision Transformers pre-trained using self-supervised, contrastive methods. Phikon (Filiot et al., 2023) was one of the earliest FMs, reaching state-of-the-art performance on several histopathology benchmarks. This model was trained with masked image modeling using iBOT (image BERT pre-training with Online Tokenizer) (Zhou et al., 2021). Among recent FMs, Virchow2 (Zimmermann et al., 2024) was considered the most robust FM by de Jong et al. (2025) and performed on par with other recent FMs in a study by Campanella et al. (2025). Virchow2 was trained using DINOv2 (Oquab et al., 2023): a mix of iBOT and DINO (self-distillation with no labels) (Caron et al., 2021). In this work, we evaluate the robustness of both Phikon and Virchow2 under domain shifts and explore a possible strategy to improve their generalization.

Given the challenge of domain shift and no available target domain during training, the problem falls within the scope of domain generalization. Various domain generalization techniques have been proposed to improve model robustness, including transfer learning and data augmentation (Zhou et al., 2022). In computational pathology, data augmentation such as stain color augmentation is commonly used to mitigate the effects of domain shift (Tellez et al., 2019). It works by using photometric and geometric augmentations in image space to synthetically simulate new domains, and has been shown to improve the generalizability

of deep neural networks for pathology (Tellez et al., 2019; Jahanifar et al., 2025). Pohjonen et al. (2022) argues that strong augmentations are needed to improve model robustness. These techniques are applied when pre-training or training the encoder.

These domain generalization approaches rely on tuning the encoder, which poses a significant limitation when working with FMs, which are typically very large and expensive to train or adapt. Re-training or fine-tuning these models for each new task or deployment is impractical. Moreover, FMs are usually trained on huge proprietary datasets that are not publicly available. Re-training on smaller datasets might make them forget important discriminatory information learned from the original data. As a result, traditional domain generalization techniques are not well suited to make the output of FMs in histopathology more robust. New strategies are needed to improve robustness without full-scale re-training. Our work aims to address this limitation by using an efficient, adapter-based tuning method tailored to large FMs. Specifically, we propose low-rank adapters to fine-tune FMs without changing their weights, using augmentations to simulate domain shifts.

## 3 Methodology

A possible solution for fine-tuning large FMs with limited data and resources is Low-Rank Adaptation (LoRA) Hu et al. (2022). LoRA is a fine-tuning technique that, instead of updating a full weight matrix, learns a weight update matrix $\Delta W$, which is reparametrized as a low-rank decomposition $\Delta W = AB$. The weight update matrix is injected into each dense layer in the network, and contains the only weights updated during training. The resulting weights are given by

$$W' = W + \Delta W = W + AB.$$

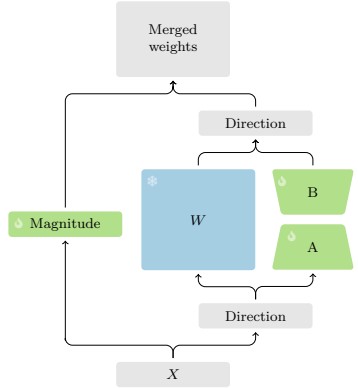

**Figure 1.** We update the model $X$ by introducing a DoRA adapter ($A$, $B$, and the magnitude) (Liu et al., 2024) that is trained to simulate the corresponding fine-tuning of $X$.

A later development of LoRA known as DoRA (Weight-Decomposed Low-Rank Adaptation), uses further weight decomposition to improve training stability and the learning capacity of the parameter-efficient fine-tuning (Liu et al., 2024). This is done by decomposing the weight matrix into magnitude and direction components, and applying LoRA to the directional component as shown in Figure 1. This extension has shown improved robustness over training parameters, and overall improved performance compared to LoRA when used on larger vision models (Liu et al., 2024). We therefore explore the use of DoRA to tackle domain generalization issues in FMs.

We follow prior work in domain generalization for histopathology by training with augmentations. We use Caron et al.'s (2021) DINO framework to tune the FMs in a self-supervised way. DINO has been shown to learn powerful visual representations that transfer more effectively than those trained with supervision (Caron et al., 2021), making DINO suitable for our goal of improving robustness of learned features. We follow Pohjonen et al.'s (2022) work and apply strong augmentations to the DINO views, including full variations in color hue, and high variability in brightness, contrast, saturation and Gaussian blur. Consistent with Zimmermann et al.'s (2024) work, we avoid heavy changes of aspect ratios,

as this may distort the biological meaning of cells and other structures in histopathology images. We also omit solarization augmentations, as this may generate ineffectual color profiles that are highly unlikely to occur as part of domain shifts in this field (Zimmermann et al., 2024; Faryna et al., 2021). Our method combines parameter efficient self-supervised tuning with strong augmentations to improve the robustness of FM features under domain shifts.

## 4 Experiments

We apply our method to three distinct Lung Squamous Cell Carcinoma (LUSC) datasets and analyze the resulting feature representations. We assess the robustness of FM features and investigate whether tuning FMs with low-rank adaptations and strong augmentations can reduce the impact of domain shifts in histopathology.

**Datasets.** We consider three datasets of LUSC images in our experiments: The Cancer Genome Atlas (TCGA), University Hospital of North Norway (UNN), and University Medical Center Mainz (Mainz). TCGA is a public dataset, and is part of the training data for the Phikon model (Weinstein et al., 2013; Filiot et al., 2023). UNN and Mainz are private datasets collected from hospitals in Norway and Germany, respectively. Although these two datasets were scanned at the same lab, staining and tissue pre-processing was performed locally at each

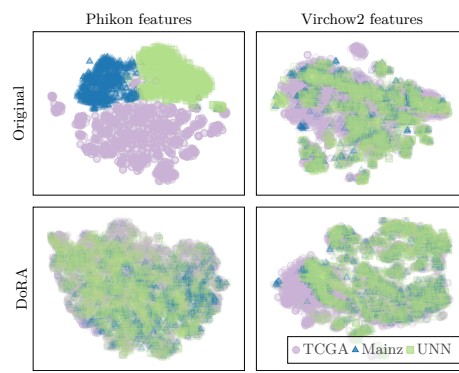

**Figure 2.** t-SNE plots for Phikon and Virchow2 original features (top) and DoRA adapted features tuned on UNN (bottom). The plots show a more clear clustering for the original features, suggesting these features are more dataset dependent. This is an undesired quality for domain generalization.

hospital. The TCGA LUSC subset includes 464 patients and 498 WSIs. The internal datasets are smaller, with UNN comprising 289 patients and WSIs, and Mainz has 232 patients and 233 WSIs.

### 4.1 Domain shifts in FM features

In this section, we analyze the effect of domain shifts across three datasets on the features generated by the original FMs. Figure 2 shows t-SNE plots of features from 10 random image patches (tiles) from each WSI, produced by Phikon and Virchow2. t-SNE plots do a non-linear dimensionality reduction that preserves local structures, such that similar input datapoints should be close together in the reduced space. The clear clustering of tile features by dataset in the Phikon feature space suggests that features from the same dataset are more similar to each other than to those from different datasets; even though the underlying biological characteristics may be similar. This highlights limited domain generalization of the features. This clustering is less apparent for Virchow2 features, but this does not necessarily indicate that the features are robust across domains (see below).

To further evaluate the robustness of the FM features, we train models to predict overall survival using the features as input to the ABMIL (Ilse et al., 2018) algorithm and compare the performance of the features on internal and external data. This ABMIL head is trained

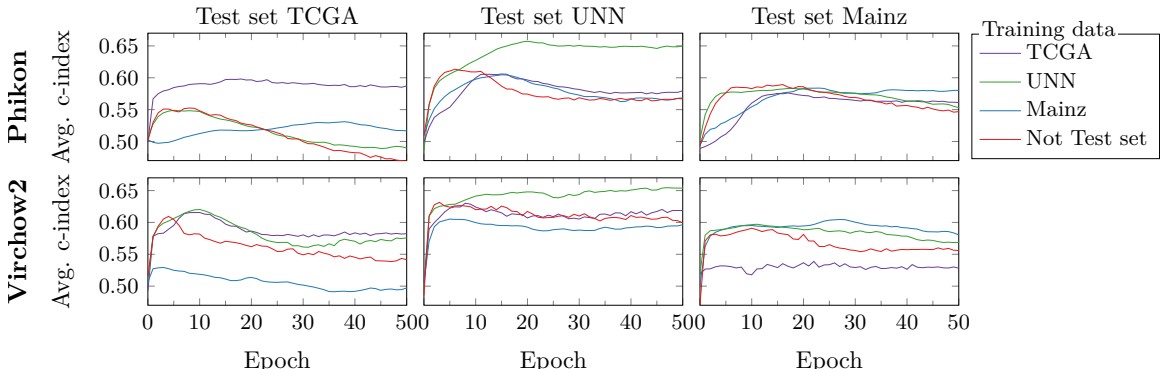

**Figure 3.** Phikon (top) and Virchow2 (bottom) performance evaluated on internal and external datasets. The performance drops when evaluated on external data, suggesting robustness issues. The "Not Test set" refers to training on the union of the two datasets not used as "test set."

for 100 epochs with a Cox loss (Katzman et al., 2018), with constant learning rate of 0.0002, and batch size 64. We use a weighted random sampler to load the training data, accounting for dataset imbalances in disease-specific survival events. We compare the performance of models in internal (ABMIL head trained and tested on the same dataset) versus external (trained on different datasets than tested on) evaluation settings. We conduct internal evaluations using 5-fold cross-validation. Each evaluation was repeated 20 times, and we report the average concordance index (c-index) (Harrell et al., 1982) across these runs. The results for Phikon and Virchow2 features are shown in Figure 3. For Phikon and test sets TCGA and UNN, we observe a considerable drop in performance when features are evaluated on external datasets. This drop is less visible when testing on Mainz, where models trained on TCGA and UNN achieve results comparable to those trained directly on Mainz. However, this may be due to the larger combined training set rather than generalization properties. For Virchow2 features, we also observe a drop in performance for external evaluation. Since robust features should generalize well across input domains, this observed drop confirms that both Phikon and Virchow2 struggle with domain robustness in survival prediction.

### 4.2 DoRA adapter for robustness

In this section, we train DoRA adapters for the Phikon and Virchow2 FMs and compare the resulting features with the original features. For each FM, we train three DoRA-tuned models, each using one of the datasets: TCGA, UNN, or Mainz, as the tuning set. The DoRA adapters are placed within each Vision Transformer block, and trained with a LoRA rank of 16, dropout rate 0.1, and for 100 epochs. We follow training hyperparameters from the original DINO paper (Caron et al., 2021), except for a reduced batch size of 4 for Phikon and 1 for Virchow2, due to GPU limitations. We evaluate the features using t-SNE visualizations, dataset clustering metrics, and K-Means clustering metrics. We also visually inspect WSI tiles in the K-Means clusters.

Figure 2 (bottom) shows that there is no longer a clear clustering of Phikon DoRA features according to dataset. This suggests that the DoRA-tuned Phikon model captures less dataset-specific information, as features from different datasets are closer in distance

**Table 2.** K-means feature clusters' purity scores with standard deviations of Phikon and Virchow2 with and without DoRA-tuning. Best results for each base FM is highlighted.

| FM | Phikon | Phikon + DoRA | | | Virchow2 | Virchow2 + DoRA | | |
|---|---|---|---|---|---|---|---|---|
| Tuning set | None | TCGA | UNN | Mainz | None | TCGA | UNN | Mainz |
| Purity score ↓ | $0.986 \pm 0.002$ | $\mathbf{0.567 \pm 0.002}$ | $0.596 \pm 0.002$ | $0.594 \pm 0.002$ | $0.632 \pm 0.007$ | $0.549 \pm 0.005$ | $\mathbf{0.501 \pm 0.003}$ | $0.573 \pm 0.005$ |

in the new t-SNE plot, compared to the original Phikon features at the top of Figure 2. We briefly experimented with weaker augmentations during DoRA training and observed less mixing of datasets in the Phikon t-SNE plots, suggesting an effect from the strong augmentations. Future work could experiment more with different augmentations. For Virchow2, there is little visible difference between the original and tuned features.

**Per-dataset clustering.** If we consider each of the three datasets as a cluster, we can use classical cluster metrics to evaluate how features from different datasets are spread out in features space. Table 1 displays the results of three cluster metrics, which are calculated using 100 random tiles from each WSI. The Silhouette score (Sil.) decreases for all tuned versions of the FMs, meaning these tile features fit less well within their dataset cluster. This indicates that the tuned models produce tile features that are less similar to other tiles in the

**Table 1.** Cluster evaluation of per-dataset clusters. We show baseline (non-tuned) Virchow2 and Phikon feature clusters (top row of each FM), and tuned with DoRA on each dataset (bottom 3 rows of each FM). Best results for each metric and base FM are highlighted.

| FM | Tuning set | Sil. ↓ | CH ↓ | DB ↑ |
|---|---|---|---|---|
| Phikon | None | 0.055 | 8990 | 2.9 |
| Phikon + DoRA | TCGA | −0.039 | 1956 | **12.5** |
| | UNN | **−0.043** | **1468** | 9.1 |
| | Mainz | −0.028 | 1501 | 7.3 |
| Virchow2 | None | 0.014 | 1734 | 7.8 |
| Virchow2 + DoRA | TCGA | 0.006 | 802 | 12.7 |
| | UNN | **0.004** | **799** | **15.7** |
| | Mainz | 0.006 | 1031 | 11.2 |

same dataset, compared to tiles from other datasets, suggesting that the new tile features capture less dataset differences and more of other differences, which could be biological differences. The Calinski-Harabasz (CH) index also decreases for all tuned versions of the FMs, meaning a lower ratio of between-clusters to within-clusters dispersion. This suggests reduced separation between features from different datasets and more separation between features from the same dataset, which is preferable for domain generalization. The Davies-Bouldin (DB) index increases with at least 149% for Phikon tuned models and 44% for Virchow2 tuned models. This means tuned dataset clusters are less compact and overlap more, which is what we desire for improved domain generalization because features from different datasets are then less separable. The consistent improvement in clustering metrics for the tuned models suggests that the DoRA adapters reduce the dataset-dependent information in the FM features compared to the original FMs.

**K-Means feature clustering.** Here, we perform K-means clustering on FM features and analyze the resulting clusters to evaluate the quality of the features. Unlike per-dataset cluster evaluations, this sheds light on which features are close in the learned space, and how different datasets and biological characteristics are distributed across the clusters.

We perform K-Means clustering on 100 random tiles per WSI, which splits the tile features into 200 different clusters. In preliminary experiments, we used 1000 tiles, which gave similar purity scores. Due to time constraints, we use 100 tiles in these experiments. First, we consider the cluster purity scores: the proportion of tiles from the most common

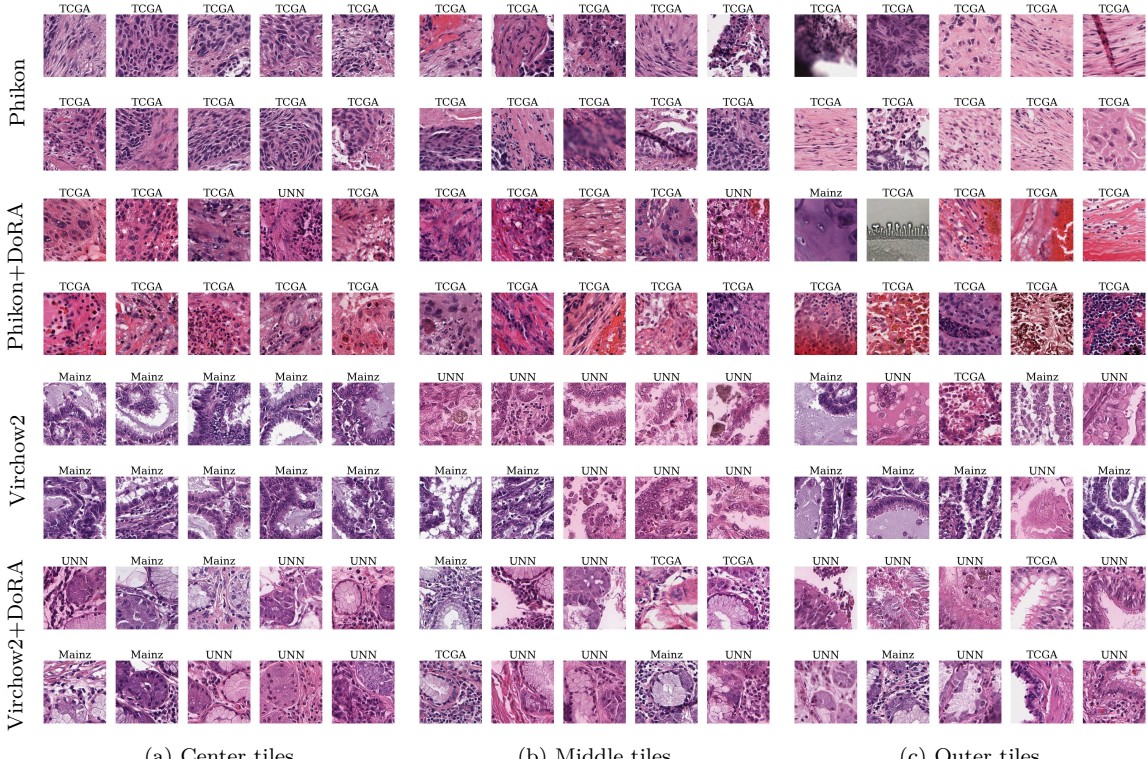

      (a) Center tiles          (b) Middle tiles        (c) Outer tiles

**Figure 4.** Tiles that are close to the a cluster center (left), have a medium distance to the center (middle), and are far from the cluster center (right) for Phikon and Virchow2 with and without DoRA-tuning.

dataset within each cluster, averaged across all clusters and over 20 repetitions of the K-means algorithm with different random initializations. The optimal purity score in our case is $\frac{1}{3}$, and the worst score is 1. The results are shown in Table 2: the DoRA tuned versions of the FMs all improve the purity score. Calculating $p$-values using a two-sample $t$-test comparing a base FM to a corresponding DoRA training, all tests give $p < 0.0001$ meaning the purity scores for tuned models are significantly different from their baseline. Since all datasets contain the same cancer subtype, it is desirable that the clusters are able to gather tiles across datasets in the same clusters. This may suggest that the tuned models are able to capture more biological similarity as opposed to dataset similarity, indicating more robust feature representations.

To further explore the difference of the K-Means clustering for the different models, we visually inspect tiles in different clusters and observe some trends. Figure 4 shows tile examples for Phikon, Phikon+DoRA (tuned on UNN), Virchow2, and Virchow2+DoRA (tuned on UNN) clusters. More examples can be found in Appendix A. The tuned clusters display greater variation in tile color profiles, particularly near the cluster centers. In contrast, center tiles from most Phikon and many Virchow2 clusters tend to have very similar color schemes, suggesting that these models encode more color-specific information in their features. This is undesirable since domain shifts in histopathology often include staining or scanning differences resulting in different color schemes of the WSIs. Tiles near the cluster centers are considered to be similar according to the learned feature space of

a model. Thus, compared to the non-tuned models, the DoRA-tuned models appear to cluster together tiles from different datasets more based on characteristics such as tissue type, and less by color (Figure 4 and Appendix A).

Considering biological similarities within the clusters, all Phikon-based models appear to have a quite large variance of tissue types within most clusters, even though tiles near the cluster centers often appear to be biologically similar. For Virchow2, some clusters do very well at capturing only similar biological tiles within the same cluster (Figure 4 and Appendix A). This is also something we find in many DoRA-tuned Virchow2 clusters, which may appear to create clusters with even greater biological similarity that include more tiles from different datasets, which reflects the improved dataset purity observed earlier (Table 2).

We also observe that low-quality artifacts such as blurred, empty, or very dark tiles are placed in different clusters for the different models (examples in Appendix Figure A.8). Phikon clusters typically have these tiles in clusters with normal cluster centers (Figure 4). For DoRA-tuned Phikon, some small clusters tend to group these artifact tiles together, reducing their presence in other clusters. Both Virchow2 and DoRA-tuned Virchow2 produce some clusters dedicated entirely to artifact tiles, but the DoRA-tuned version also distinguishes between the different types of artifacts, and generates more clusters for them. This could be because biologically similar tiles from different datasets are more frequently clustered together in the DoRA-tuned Phikon and Virchow2 compared to the original Phikon and Virchow2, which reduces the purity score and makes it possible with additional clusters to represent different types of artifacts.

Overall, these results indicate that DoRA-tuning with strong augmentations improves cluster quality by encouraging invariance to staining and scanner differences while keeping biological information, thereby enhancing robustness of the models' features. This raises a question of whether current pre-training strategies for foundation models in histopathology are sufficient, or if performance could be improved either by incorporating stronger augmentations during pre-training, or by adding a post-training step similar to our DoRA adaptation with heavy augmentations.

## 5 Conclusion

In this work, we evaluated the robustness of FM features in computational pathology, and found that current models exhibit non-robust behavior. The FMs we tested performed worse on external test data than on training data, and their features encoded dataset-specific information, limiting generalizability. To address this, we explored training a DoRA with strong augmentations for the FMs, which showed promising improvements in feature robustness across domains, as demonstrated by clustering evaluations and visual inspections. Our results highlight the importance of testing robustness for models, even when trained on large-scale data. Future work should assess more FMs across diverse test datasets, and explore additional adaptation strategies. Establishing a benchmark for robustness in histopathology FMs would help guide progress in the field. Strengthening the reliability of FMs is important to build clinical trust and realize their full potential in computational pathology.

## Acknowledgments and Disclosure of Funding

The results shown here are in whole or part based upon data generated by the TCGA Research Network: https://www.cancer.gov/tcga. This work was supported by the authors' institutions and a grant from the Research Council of Norway (project number 309439).

## Appendix A. Additional Results

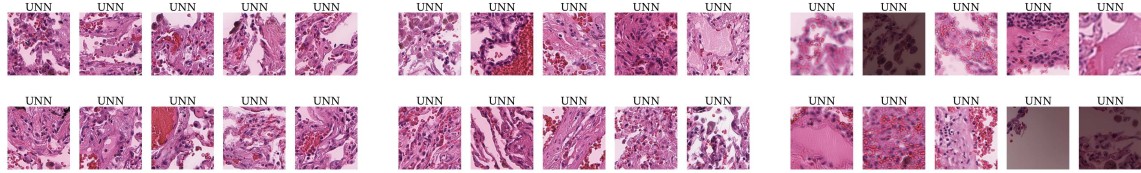

**Figure A.1.** A Phikon cluster. This cluster contains tiles from only UNN and Mainz. Several tiles far from the center have artifacts: many are blurred and one contains very little tissue. Most tiles have a very similar color profile.

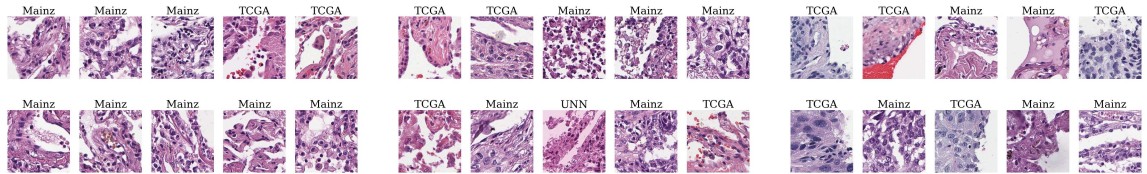

**Figure A.2.** A Phikon+DoRA tuned on TCGA cluster. The center tiles (left) consist of purple-hued Mainz tiles and more pink-hued TCGA tiles. The cluster contains tiles from all three datasets.

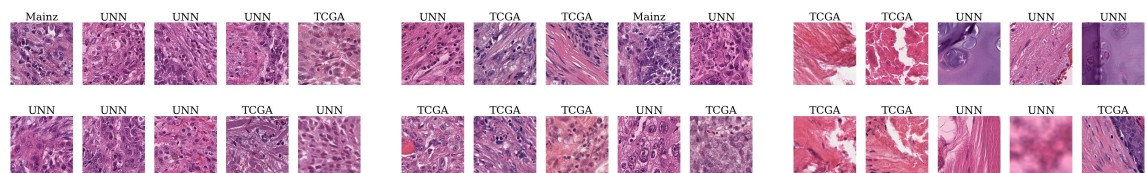

**Figure A.3.** A Phikon+DoRA tuned on Mainz cluster. This cluster contains tiles from all three datasets. We notice biological similarities among tiles near the cluster center (left), but more differences when we also consider tiles far from the center (right). There are artifact tiles present in the cluster far from the cluster center (right).

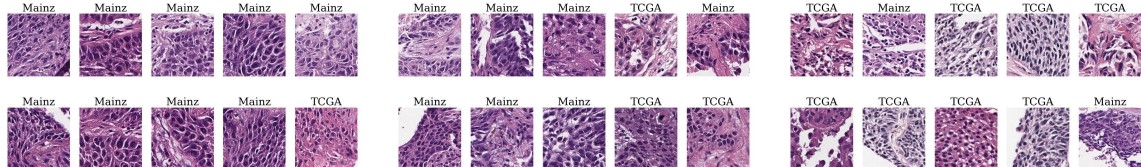

**Figure A.4.** A Virchow2 cluster. We notice biological similarities among tiles near the cluster center (left), but more differences when we also consider tiles further from the center (middle + right).

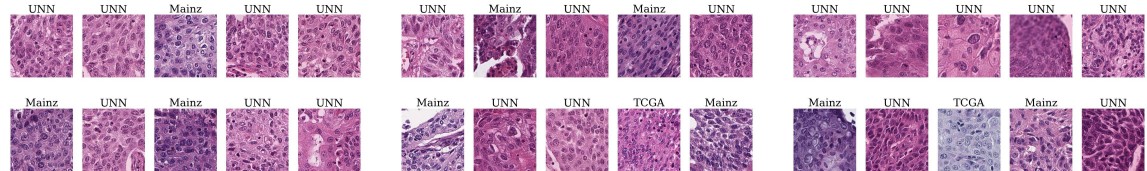

**Figure A.5.** A Virchow2 cluster. We notice biological similarities among tiles throughout this cluster, as well as some color variations between center tiles.

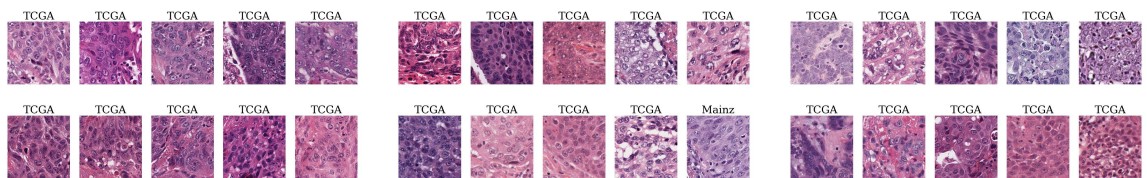

**Figure A.6.** A Virchow2+DoRA tuned on Mainz cluster. The cluster contains tiles from all three datasets. We notice biological similarities among tiles throughout this cluster, as well as large color variations between the tiles.

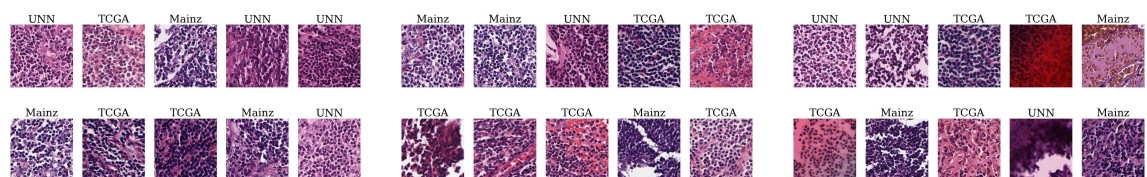

**Figure A.7.** A Virchow2+DoRA tuned on TCGA cluster. We notice biological similarities among tiles throughout this cluster, as well as large color variations between the tiles.

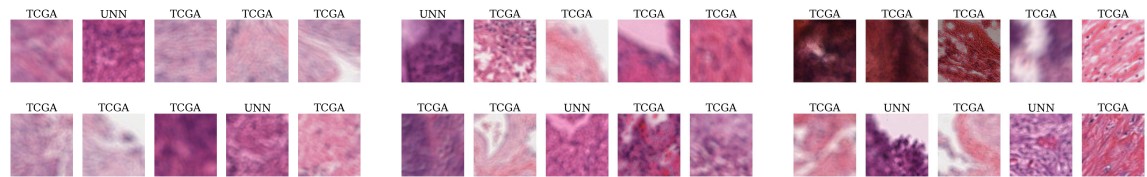

**Figure A.8.** A Virchow2+DoRA tuned on UNN cluster: all tiles appear heavily blurred. This is an example of a cluster with artifact tiles.

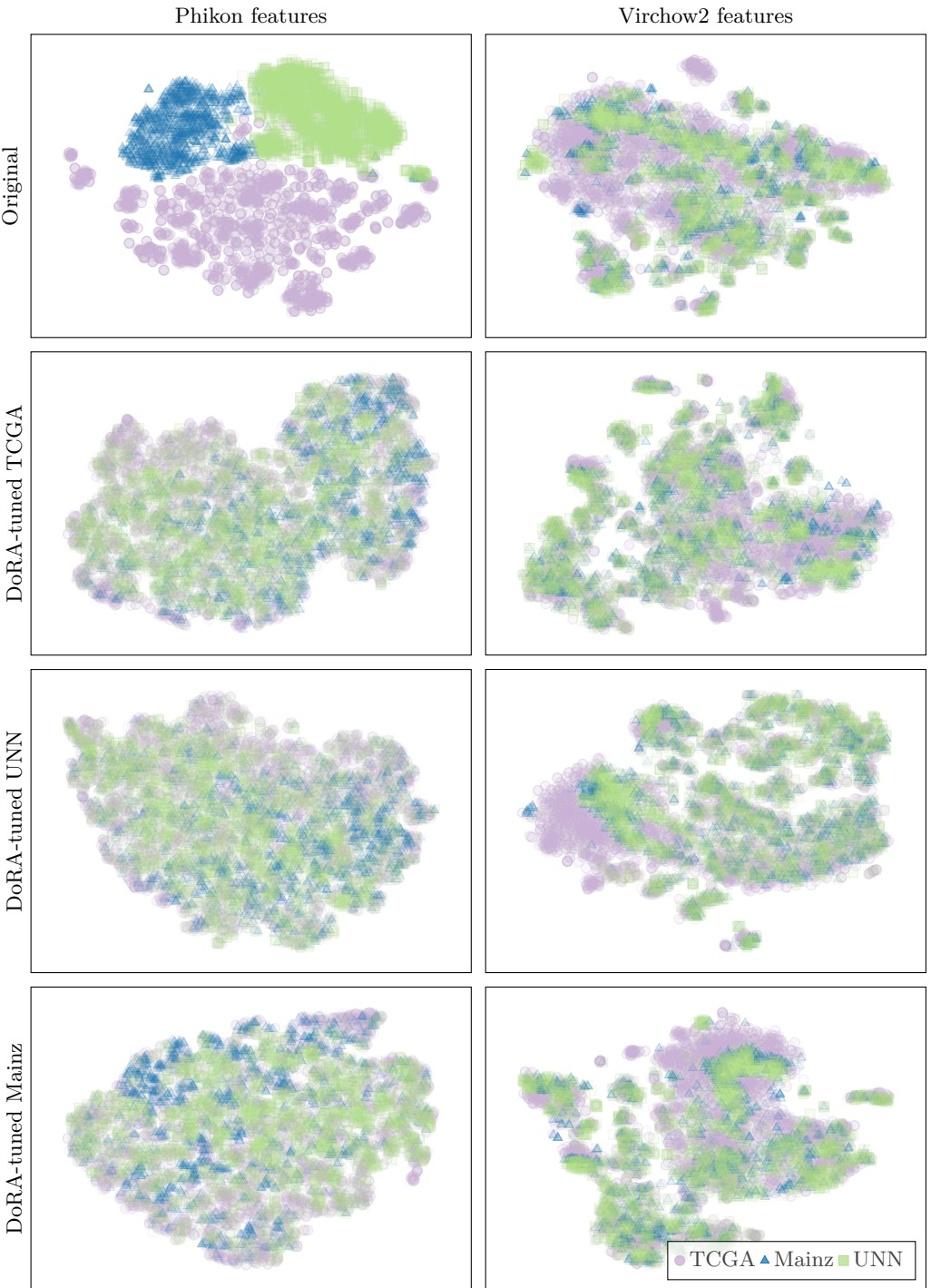

**Figure A.9.** t-SNE plots for Phikon and Virchow2 original features (top) and DoRA adapted features trained on the three different datasets (bottom three). For all models, we extract features for all features and plot 10 random tile features from each WSI. The plots show a more clear clustering for the original Phikon features, suggesting these features are more dataset dependent. The different DoRA-tuned Phikon features all show less apparent dataset clustering.

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
