# OpenReview forum: "Low-Rank Adaptations for increased Generalization in Foundation Model features"
_MICCAI.org/2025/Workshop/COMPAYL — COMPAYL 2025_

### Official Review · Reviewer_oAmM · 2025-07-12
**Comparison of two vision models fine-tuned with DoRA on three datasets**

**Rating:** 4
**Confidence:** 5

**Review:**

This work addresses questions about generalization of features extracted from vision-only foundation models for pathology.
The authors consider two publicly available models (Phikon and Virchow2) and tested their performance on three datasets of H&E-stained lung squamous cell carcinoma surgical resections.
Features are analyzed in multiple ways, both quantitavely and qualitatively.
"Baseline" features are used to predict survival via an ABMIL approach. Mutiple configurations with training data from 3 cohorts are used and tested on data from those cohorts. This showed poor generalization of features for survival across cohorts.
The authors then trained models with DoRA adapters and strong color augmentation and showed increased generalizability though several quantitative metrics: reduction of silhouette score, reduction of purity score for per-dataset clusters made with K-Means.

#### PROs
* The topic covered by this paper is relevant, evidence is needed to better understand the potential benefits of foundation models in pathology.
* The use of three datasets, including external data for the considered models, allows to measure the generalization of features in an unbiased way.
* Code will be made available, which is good for reproducibility (at least for the TCGA part); it would be great if private datasets would be also made available in the end.

#### CONs
* Unclear why Phikon and Virchow2 were chosen and not other models. From the text, it seems that Phikon was chosen because it was one of the first models to be released, and Virchow2 because is among the best in the Pathbench benchmark, but other models could have been tested. This is the trend that was followed in recent work as for example Campanella et al., 2025 (https://doi.org/10.1038/s41467-025-58796-1). The choice of the used models should be better justified in the paper, in relation to the main contribution of this manuscript (i.e., if the focus is more on the contribution of the DORA approach).
* K-Means clustering was performed on 100 random tiles per WSI. Why was this choice made? Is it based on previous work? Also unclear if random (uniform?) sampling is sufficiently representative here.
* The qualitative analysis of patches in relation to clusters is nice, but using a dataset with labelled patches, or with manual annotations of tissue compartments, could have also provided quantitative insights. This is an aspect that I am missing in the paper, prediction of survival is relevant but is a very complex task, a benchmark more closely related to morphological features (like patch classification) could have shed more light on the advantages of the proposed DoRA approach. The authors might consider this as future work, as now one can only appreciate the (dis)similarity in morphology across patches, but for non pathologists this might be difficult to grasp.
* The focus of this work is on DoRA-tuned models, with strong color augmentation. Unclear why this specific DoRA approach is used, whether alternatives could be considered for model fine-tuning, and how much the color augmentation or the specific DoRA fine-tuning approach contributes to the measured performance.
* When training to predict overall survival, the authors should report the experimental setting, in particular the type of loos that was used and how training mini-batches were constructed (i.e., the distribution of of survival within a mini-batch), as these are known to be relevant hyper-parameters when training ABMIL to predict outcome.

---

### Official Review · Reviewer_o6bD · 2025-07-14
**Low-Rank Adaptations for Enhanced Generalization in Foundation Model Features using Deep Learning**

**Rating:** 2
**Confidence:** 5

**Review:**

The authors propose Low-Rank Adaptations for Enhanced Generalization in Foundation Model Features using Deep Learning. They provide a comprehensive background on the problem and clearly articulate why improved generalization is necessary for foundation models. The paper reviews relevant work on foundation models, alternative methods and addressing the significant computational burden of foundation model training. This research proposes DoRA, a variant of LoRA (Low-Rank Adaptation) designed to enhance the generalizability of foundation models. To demonstrate that foundation models often fail to generalize across different centers, they conduct experiments evaluating survival performance consistency across institutions. Their experimental validation used three lung cancer datasets: (i) The Cancer Genome Atlas (TCGA) (ii) Hospital of North Norway (UNN), and (iii) University Medical Center Mainz (Mainz). Their evaluation using K-means clustering revealed that models trained with DoRA do not preserve color-specific characteristics and cluster contains patches from cross-centers, indicating preserving biologically relevant features during the adaptation process.
Weaknesses:
i.	The authors show that Virchow demonstrates superior generalization compared to Phikon. However, given that stronger baselines like UNI exist, it would strengthen the evaluation to include comparisons with these state-of-the-art foundation models alongside Virchow.
ii.	The manuscript would benefit from a more detailed explanation of the DoRA integration within the network architecture. Please clarify whether DoRA blocks are inserted after/within each ViT block, or at other stages to help readers better understand the architecture.
iii.	Authors presented that existing models struggle with cross-center generalization by training a model and reporting c-index, they didn’t include these evaluation metrics to provide a more comprehensive and fair comparison.
iv.	Please provide more detailed explanations of how "strong augmentations" specifically address cross-domain challenges, including the underlying mechanisms and rationale.
v.	The study would benefit from including downstream task evaluations to complement the K-means clustering analysis. Please provide additional explanation for using K-means approach, as clinical utility depends on both feature separability and robust performance on relevant downstream tasks.
vi.	In A.9's, the plots for Virchow appear like other methods, making it challenging to assess the effectiveness of the proposed approach. Please consider improving this visualization by visualizing the data together.

---

### Official Review · Reviewer_BmZb · 2025-07-21
**Low-Rank Adaptations for increased Generalization in Foundation Model features**

**Rating:** 4
**Confidence:** 4

**Review:**

This paper aims to address a shortcoming of digital pathology foundation models (FMs) that has become apparent recently, in that they tend to encode non-biologically relevant information quite strongly. Ideally you would like tissue to be clustered according to biology in the embedding space of model, and not according to scanner type, or particulars of the staining protocol used, etc.

The authors address this by fine-tuning the model using a type of LORA, allowing to adjust the model weights by learning a much smaller set of weights than the full weight matrix.

Strengths:

The use of a LORA is a sensible approach that has been proved to be effective in a wide variety of fine tuning tasks across other domain, and allows effective fine-tuning of large models on relatively modest computational resources and dataset sizes.

The paper is clearly written and helps address a significant current problem with pathology foundation models. While none of the methods in here are original, to my knowledge no-one has yet applied them to address this task in the digital patholgy domain and evaluated their effectiveness, so the paper is a valuable contribution regardless. The authors choose the components of their fine-tuning approach well; DORA, DINO and the strong augmentation library are all proven effective approaches.

The results presented in the paper convincingly demonstrate a reduction in clustering by non-biological factors in favor of biologically relevant differences.

Weaknesses:

One weakness I see in the experimental results is that, while their experiments clearly demonstrate nicer clustering properties of features, they do not present experiments demontrating that this translates to improved downstream prediction performance. As the authors trained a fine-tuned model on each of the 3 datasets, it would have been nice to see a comparison to some of the c-indices in figure 3 using, in the case of the experiments with single train set and external test set, the respective fine-tuned model from that train set.

A minor discussion point I also felt was missing, is the implications of these findings for foundation model training. When we are doing the original training for our FM, should we not based on this work include a post training bit of extra training with stronger augments (or just use stronger augments from the start during the training if it does not destabilize things?). The training paradigm they use in this paper (DINO) is roughly the same as the typical approach used when initially trainng these models, so it would presumably slot in just fine. Then the benefits shown here would be baked in to the foundation model from the start.

Overall I think it is a paper worth accepting; I considered giving it a strong accept but opted eventually for a weak accept due to the lack of experiment demonstrating that improved feature robustness translates to improved robustness in predictive performance.